# Copper/Zinc Ratio in Childhood and Adolescence: A Review

**DOI:** 10.3390/metabo13010082

**Published:** 2023-01-03

**Authors:** Marlene Fabiola Escobedo-Monge, Enrique Barrado, Joaquín Parodi-Román, María Antonieta Escobedo-Monge, María Carmen Torres-Hinojal, José Manuel Marugán-Miguelsanz

**Affiliations:** 1Faculty of Medicine, University of Valladolid, Avenida Ramón y Cajal, 7, 47005 Valladolid, Spain; 2Department of Analytical Chemistry, Science Faculty, Campus Miguel Delibes, University of Valladolid, Calle Paseo de Belén, 7, 47011 Valladolid, Spain; 3Science Faculty, University of Cadiz, Paseo de Carlos III, 28, 11003 Cádiz, Spain; 4Department of Chemistry, Science Faculty, University of Burgos, Plaza Misael Bañuelos sn, 09001 Burgos, Spain; 5Department of Pediatrics, Faculty of Medicine, University of Valladolid, Section of Gastroenterology and Pediatric Nutrition, University Clinical Hospital of Valladolid, Avenida Ramón y Cajal, 7, 47005 Valladolid, Spain

**Keywords:** chronic diseases, zinc-to-copper ratio, inflammation, oxidative stress, nutritional status

## Abstract

Both copper (Cu) and zinc (Zn) are crucial micronutrients for human growth and development. This literature review covered the last five years of available evidence on the Cu/Zn ratio in children and adolescents. We searched PubMed, Web of Science, Google Scholar, Cochrane Library, and Science Direct for publications between 2017 and 2022, especially in English, although publications in other languages with abstracts in English were included. The main terms used were “copper”, “zinc”, “copper-zinc”, and “zinc-copper” ratios. Cu and Zn determinations made in blood, plasma, or serum were included. This review comprises several cross-sectional and case–control studies with substantial results. The bibliographic search generated a compilation of 19 articles, in which 63.2% of the studies mostly reported a significantly higher Cu/Zn ratio, and 57.9% of them informed significantly lower levels of Zn. We conclude that children and adolescents with acute and chronic conditions are at greater risk of developing elevated Cu/Zn ratios, related to altered nutritional, infectious, and inflammatory status.

## 1. Introduction

The prevalence of chronic diseases is increasing rapidly, especially affecting younger people, reaching more than 10–20% of the child population [1,2]. Currently, we know that many chronic diseases begin in childhood, and early diagnosis and treatment mean that more than 90% of patients with chronic conditions or disabilities, such as cystic fibrosis (CF) [3], can survive beyond the age of twenty [2]. The obesity epidemic is a clear example of a chronic condition that is becoming more frequent at younger ages and that is related to other non-communicable diseases, such as glucose intolerance (or pre-diabetes), cardiovascular diseases (CVDs), diabetes mellitus (DM), metabolic syndrome (MetS), among others [2]. In addition, it can act as an independent risk factor for the development of type 2 diabetes, insulin resistance, dyslipidemia, cancer, etc. [2,4]. In overweight and obese children, high levels of copper (Cu) [5,6] with significantly lower levels of zinc (Zn) [7] have been reported.

Additionally, a Cu imbalance, whether hereditary or acquired (deficiency, overload, or maldistribution), can cause or aggravate certain diseases, such as Menkes disease, Wilson’s disease, and neurodegenerative diseases, including Parkinson’s disease (PD) [8,9], Alzheimer’s disease, Huntington’s disease, prion disease, systemic lupus erythematosus [8], anemia, and cancer [10]. Exposure to toxic copper is associated with an increased risk of CVD and coronary heart disease, MetS [5,10], heart failure, and ischemic stroke in epidemiological studies [11]. Wilson’s disease is an autosomal recessive disorder of Cu deposition in the liver, caused by an alteration in the Cu transporter gene, ATP7B. An early detection and treatment are essential to prevent lifelong neuropsychiatric, hepatic, and systemic disabilities [12]. Menkes disease, occipital horn syndrome (OHS), and ATP7A-related distal motor neuropathy are disorders caused by the alteration in ATP7A, an X-linked gene encoding a Cu-binding ATPase. Menkes disease and OHS are characterized by low Cu levels in some tissues due to impaired intestinal Cu uptake, accumulation in other tissues, and reduced activity of Cu-dependent enzymes such as dopamine-beta-hydroxylase and lysyl oxidase [13].

It is important to highlight that Cu is a crucial micronutrient for the proper functioning of the human organism [14]. In adult populations, several studies show that the Cu/Zn ratio is a better predictor of disease severity and/or mortality than Cu levels. It can be used as a prognostic and predictive factor for screening diagnosis of Zn-induced Cu deficiency and in a series of pathological and pre-pathological conditions, degenerative diseases, lymphoproliferative disorders, various types of cancer, and all causes of mortality in the elderly. However, there are also studies of the Zn/Cu ratio in children, for example, in the autism spectrum disorder (ASD). [2,3]. There are few studies on Cu levels in childhood and adolescence and even fewer on the Cu/Zn ratio. Therefore, the main aim of this narrative review is to summarize and synthesize the advances in knowledge of the last five years on the Cu/Zn or Zn/Cu ratio in children and adolescents.

## 2. Materials and Methods

The present literature review was conducted in accordance with the principles of the Declaration of Helsinki on data management. 

We conducted an exhaustive review of the literature on Cu levels and the Cu/Zn ratio in children and adolescents. The research was conducted on PubMed, Web of Science, Google Scholar, Cochrane Library, and Science Direct, as follows: (1) the keywords considered were: zinc and copper levels, copper to zinc ratio or zinc to copper ratio); (2) use of the Boolean variable (“AND”); (3) Cu and Zn levels determined in plasma, serum and whole blood samples; (4) articles published from 2017 to 2022; (5) ages from birth to 19 years (infants, children, and adolescents); (6) main language English and articles in other languages with abstract in English; (7) no type of acute or chronic disease were excluded; 8) studies conducted in adults, women (females), men (males), pregnant women, and the elderly were excluded from the search.

The following information will be collected from the articles: (1) type of population studied: specific disease; (2) type of study: cross-sectional study (CSS), case–control study (CCS), randomized trial study (RTS); (3) year of publication; (4) the number of cases (the number of subjects in the control group is not shown); (5) age: range or mean and standard deviation (SD), age range: children (≤10 years) and adolescents (10–19 years) according to Tanner’s sexual development; (6) levels of Cu, Zn, and Cu/Zn and Zn/Cu ratios of cases and controls, the results will be shown mainly in ug/dL, (it will be written in the first column if the determinations appear in other units); (7) significantly different results will be highlighted (* < 0.05, ** < 0.001); (8) normal ranges: serum Cu between 70 µg/dL and 140 µg/dL, serum Zn between 70 and 120 µg/dL. Cu/Zn ratio between 0.7 and 1.0 [15], as a biomarker to assess inflammatory and nutritional status and adverse clinical outcomes, and Zn/Cu ratio < 4.0, associated with increased susceptibility to bacterial and viral infections [16].

## 3. Results

The bibliographic search generated a compilation of 19 articles, which describe the levels of Cu, Zn, and the Cu/Zn or Zn/Cu ratios, determined in whole blood, serum, and plasma samples, in children and adolescents with specific pathologies and published in the last five years (Table 1). 

Only 19 studies out of the 26 articles scanned (6822 subjects) reported Cu/Zn or Zn/Cu ratios. From newborns to individuals 20 years of age, 2004 participants took part in 12 CCS, 6 CSS, and one RTS. Participants were children and adolescents with specific diseases. More studies were carried out (11 articles) during 2020 and 2021 than the previous years (8 articles from 2017 to 2019). Most of the Zn and Cu determinations were made from serum (12), one of which reported levels serum Cu and plasma Zn, in cord blood (1), in plasma (3) and whole blood (3).

Although normal micronutrient values may differ geographically, the disease group was always compared with a control group from the same region. It was observed that 57.9% of the studies (11/19 articles) reported significantly lower levels of Zn in the disease cases than in their controls. Of these 11 studies, 36.4% described normal Cu levels, 27.3% informed significantly low Cu levels, and 36.4% reported high Cu levels.

Furthermore, 45.4% (5/11 articles) described a significantly higher Cu/Zn ratio in cases than in controls. A total of 27.3% reported a significantly lower Zn/Cu ratio (high Cu/Zn ratio), only 27.3% reported a low Cu/Zn ratio, and there were only significant results in two articles.

In two studies, representing10.5% (2/19 articles), adequate levels of Zn and Cu, and Zn/Cu ratio (>4.0) were reported. Only 9% (1/19 articles) described significantly high Cu levels and Cu/Zn ratio and normal Zn levels. Of the studies, 26.3% (5/19 articles) were without a control group. In one study, a significantly higher Cu/Zn ratio is observed; another one showed a Cu/Zn ratio within normal ranges; and in the other three, the Cu/Zn ratio was high based on the cut-off points used. Therefore, 63.2% of the studies mostly reported a significantly higher Cu/Zn ratio.

The main findings are commented below in the discussion.

## 4. Discussion

To the best of our knowledge, this is the first literature review carried out with the main objective of summarizing and synthesizing the advances in knowledge of the last five years on the Cu/Zn or Zn/Cu ratio in children and adolescents. The bibliographic search generated a compilation of 19 articles, in which 63.2% of the studies mostly reported a significantly higher Cu/Zn ratio, and 57.9% of the studies reported significantly lower levels of Zn.

### 4.1. Zinc and Copper Levels and Copper to Zinc Ratio

We must point out that both Cu and Zn play a fundamental role in the life cycle of all living beings. In humans, both trace elements are essential for reproductive health [33]. Zn is used in the human body for normal growth and development from the time of intrauterine development [25]. The impact of Zn level was observed in a study in which women with Zn deficiency (<7.80 μmol/L) (22.4% of 1060 pregnancies) took 0.6 months longer to conceive [34], and in the Adelaide SCOPE cohort, women with lower plasma Cu levels were reported to be protected against the risk of any pregnancy complications [35]. Neonates and especially preterm infants have an increasing requirement for Cu and Zn [17]. In 41 small for gestational age newborns, cord serum Zn was significantly lower while the Cu/Zn ratio was significantly higher than appropriate for gestational age babies. The cord serum Zn correlated positively while the Cu/Zn ratio correlated negatively with the birth weight of neonates [18].

In a nationally representative sample of the US population, from the National Health and Nutrition Examination Survey (NHANES) from the period of 2011–2014, 1427 participants aged 6–19 years had a normal mean serum Cu level of 121.7 ± 23.5 μg/dL. These levels were significantly higher in children (115.6 ± 27.2) than in adolescents (108.8 ± 29.3 μg/dL), both within normal limits [36]. These results are similar to a study conducted in a series of children and adolescents with chronic diseases, where the serum Cu level in normal ranges was also significantly higher in children (128 μg/dL) than in adolescents (106 μg/dL) [2]. These results may show the higher Cu requirement at earlier ages. Cu plays a crucial role in the development, maturation, and proper functioning of the immune system [37]. It exists as a transition metal that forms monovalent and divalent cations: Cu+ is the reduced cuprous form and Cu^2^+ is the oxidized cuprous form of Cu [38]. Cu, as a pro-oxidant, promotes the harmful effects of free radicals [39], and as an antioxidant, it eliminates free radicals and neutralizes their potentially harmful effects [40].

### 4.2. Copper to Zinc Ratio in Acute Infections

Acute infections alter micronutrients metabolism, while deficiencies increase infection risks. Infections reduce serum Zn while Cu and ceruloplasmin (CP) increase. The Cu/Zn ratio combined both alterations, regardless of gestational age. In 21 infected term and preterm newborns, Cu levels were increased in infected newborns and positively correlated with CP levels. The Cu/Zn ratio was relatively high in infected newborns and was positively correlated with C-reactive protein (CRP) levels. The odds ratio for the Cu/Zn ratio was 9.067, indicating a higher probability of infection. Hence, it constitutes a significant diagnostic biomarker for early onset infections [17]. Furthermore, acute infections are at the origin of an increase in the serum and local concentration of Cu, leading to Cu toxicity in the pathogen [41] and a decrease in serum Zn due to redistribution through the activation of inflammatory cytokines in the liver [42] and other tissues. These two responses to inflammation may feed a vicious cycle of impaired immune defense and higher infection risk, which is of particular importance especially in very vulnerable patients, such as preterm and term neonates with an immature immune system [17].

Copper has been implicated as a pro-oxidant and in the metal accelerated production of free radicals that may affect oxidative stress (OS) [43]. OS, by altering cellular homeostasis, causes a cellular response that will depend on the severity and type of aggression. There is a limiting threshold below which cells trigger protection mechanisms to ensure survival. Instead of that, if stress exceeds the threshold or if the activation of protective mechanisms fails, cells will trigger alternative signaling pathways that ultimately lead to apoptosis, necrosis, pyroptosis, or autophagic cell death [44]. A high fraction of free Cu in serum can be harmful due to its significant oxidation-reduction potential through the generation of reactive oxygen species in Fenton and Haber–Weiss-type reactions [45]. Zn serves as an antioxidant, and changes in its concentrations may impact the homeostasis of OS [43]. Cu is an essential micronutrient with antioxidant properties mediated by many redox enzymes [46]. A deranged Cu/Zn ratio may be associated with a decreased capacity of the organism to maintain or regenerate Cu–Zn homeostasis [43].

The Cu/Zn ratio has been proved to be diagnostic in several infectious disorders [47], including giardiasis or amoebiasis and tuberculosis [17]. Malaria is an important protozoan disease in developing countries [48]. Nutrition is not only vital to reduce the risk of an individual’s susceptibility to malaria infection but also improves disease prevention and treatment, as well as modifies the course of the disease, especially among children [49]. In 200 malaria-infected children (0.5–11 years) from Nigeria, the serum Cu and Cu/Zn ratio were significantly higher, and Zn was significantly lower. Serum Cu and the Cu/Zn ratio correlate positively, and serum Zn negatively correlates with malaria parasite density. Significantly higher Cu and Cu/Zn ratio and lower Zn levels may indicate OS, inflammation, and lower immune status in malaria infections [19]. Malaria is an oxidative and inflammatory disease, in which innate immune cells engulf pathogens and attempt to eliminate them by exacerbating free radical generation in their phagocytic cells through oxidative or respiratory bursts. The significantly higher Cu level may be due to the body’s attempt to neutralize these free radicals that are generated during the infection [50].

### 4.3. Cardiovascular Diseases

Inflammation is a natural defensive response, closely associated with OS. Excessive and uncontrolled inflammation causes numerous diseases, such as CVD, hepatitis, nephritis and delayed wound healing [51]. In response to growth factors, cytokines, and hypoxia, Cu expression, subcellular localization, and function are tightly regulated [46]. Cu is considered a critical cofactor for a group of cellular transporters, namely, the cuproenzymes [14]. Examples of cuproenzymes of relevance to CV physiology include Cu, Zn-superoxide dismutases (SODs) (SOD1 and SOD3), lysyl oxidase, mitochondrial cytochrome c-oxidase, and many others [46]. Recent evidence reveals that Cu transporters and chaperones play an essential role in the physiological responses of CV cells, including cell growth, migration, angiogenesis, and wound repair, being related to various CV pathophysiologies, such as hypertension (HTN), inflammation, atherosclerosis, DM, cardiac hypertrophy, and cardiomyopathy [46]. Wilson’s disease patients who had genetic Cu overload conditions have been reported and had increased risks of atrial fibrillation and heart failure [52].

Cardiovascular diseases are the leading cause of death worldwide [53]. Although in 39 children with a diagnosis of congenital heart disease (CHD: 11 with cyanotic and 28 with acyanotic CHD), which is the most common birth defect (incidence of 1.9 to 9.3/1000), there were no significant differences in blood sample for Zn, Cu, and Zn/Cu ratio, the teeth–Cu level was considerably higher in the CHD group [28]. The most prominent CV risk factors in children and adolescents are overweight, arterial HTN, and alterations in lipid and glucose metabolism [54]. It was estimated that 40 million children under 5 years of age and more than 330 million children and adolescents aged 5 to 19 years were overweight or obese in 2016 [55]. A NHANES 2011–2016 study—based on a 1242 sample of children and adolescents—showed that serum Cu was associated with increased risk of elevated blood pressure (BP) in the US population aged between 8 and 17 years old [56]. Furthermore, in order to examine the association between Cu and Zn and obesity, a cross-sectional study was performed in 173 healthy children (6–16 years). The results show that Zn and Cu were found to be significantly lower in 69 children with exogenous obesity [30]. In contrast, in 87 obese children (30 hypertensives/57 normotensive), although there was no difference between serum Zn levels, the Cu levels were significantly lower in the HTN group. There were positive correlations between BP and body mass index (BMI) and weight Z-scores and a negative correlation with Cu [57].

Reactive oxygen species (ROS) could aggravate localized tissue damage and cause chronic inflammation [51]. Excessive Cu can affect the activity of ROS [46]. The Cu/Zn ratio is recommended as a potential biomarker of nutritional status, inflammatory status, due to altered immune system and increased OS [2] as well as a predictor of mortality, such as in lung cancer [58]. Diabetes mellitus and MetS prevalence are showing a trend of becoming younger [59]. Poor glycemic control in type 1 diabetes mellitus (T1DM) usually leads to more OS, increased production of oxygen-free radicals, and more diabetic complications. In 100 T1DM, children (BMI 18.5 ± 1.7) were divided into poor and good controlled patients according to glycosylated hemoglobin (A1c %). Serum Zn and Cu were significantly lower in T1DM children, and their levels were lower in poorly controlled patients compared to good-controlled ones [60]. A reduction in insulin concentrations may result in an accumulation of Cu in the liver. If metal levels are altered in patients with DM, the management of their disease will be more complex, and T1DM patients are more likely to develop complications [61].

### 4.4. Type 1 Diabetes Mellitus and Metabolic Syndrome

Copper deficiency generates hypercholesterolemia and increased OS, which can lead to HTN [6]. In the NHANES 2011–2014 study, significant relationships were observed between serum Cu and total cholesterol (TC), A1c %, and fasting insulin. Participants with the highest Cu levels showed a 0.83% increase in serum TC, providing evidence that serum Cu is associated with TC concentrations in children and adolescents. In addition, serum Cu levels were significantly higher in subjects who were younger, girls, non-Hispanic black, and with lower incomes [36]. On the contrary, in a series of 78 children and adolescents with chronic conditions, the risk of finding altered Cu levels was higher in children and males than in adolescents and females [2]. The gender difference may be due to Cu playing a role in the action of estrogens, in turn, influencing lipid homeostasis. Genetics may be the reason why different ethnicities experience different Cu levels [36]. Likewise, a China National Nutrition and Health Survey 2010–2012 study showed that in 911 rural Chinese children (6–12 years), whole blood Cu and Zn levels were all in the normal range, and either alone or in combination had associations with MetS components—BMI, waist, triglycerides (TGs), HDL-C, fasting glucose (FG), systolic and diastolic blood pressure. Single metal analysis shows that Cu was meaningfully associated with elevated waist, low Cu, and high Zn with elevated TG, high Cu with elevated waist, and high Zn with reduced HDL-C. All the values of the Cu/Zn ratio were less than 1.0, indicating that the antioxidant mechanism in this population was normal. This ratio may be used to determine the level of OS [1].

On the other hand, a higher level of whole blood Cu leads to an increased risk of elevated waist, which was representative of obesity. Catherine et al. observed a positive relationship between serum Cu and abdominal obesity. Furthermore, the risk of elevated TGs was decreased with ascending tertiles of Cu [62]. However, there were no associations between serum Cu and TG levels among children and adolescents in the NHANES 2011–2014 study [6]. One hypothesis suggests that a key mechanism leading to MetS involves OS caused by redox imbalance [63]. Thus, Cu and Zn may be related to the development of MetS. Cu acts as an electron transfer intermediate in redox reactions [64], which has a high level of oxidation and may lead to the excessive damaging of ROS by redox reactions, affecting Zn-dependents processes [65]. Zn as a co-factor of antioxidant enzymes also plays an important role in oxidative stress and inflammation [66].

### 4.5. Myopic Patients

Another important role of Cu is its contribution in collagen synthesis and elastic fibers of collagen that form the sclera [32]. Myopia is an important health and social issue in the world today [67] and one of the most common eye pathologies [68]. It is especially worrying that an increasing number of myopia cases are observed among children. High myopia (≥6 D) may lead to vision loss due to, e.g., retinal detachment, myopic choroidal neovascularization, glaucoma, or cataract [67]. Myopia has a complex etiology and OS is one of the pathways involved in its development [68]. In 83 children (14.36 ± 2.49 years) with myopia, although there was no essential difference in serum Cu, there was a significantly lower Zn level and a higher Cu/Zn ratio. This higher ratio indicates alterations in the antioxidant mechanism and could suggest a relationship between myopia and OS. Low serum Zn in these patients may imply an association between Zn insufficiency and the development of myopia. The normalization of this ratio with Zn supplements may restore the oxidation-reduction balance and prevent myopia development [32]. Moreover, the Zn level in the hair of myopic patients was significantly higher (260 µg/g) in comparison to the control group (130 µg/g), and there was a significantly lower Cu/Zn ratio in myopic patients (0.045) compared with the controls (0.07). In these patients, an accumulation of hair Zn and insufficiency blood Zn may occur, and consequently its lesser availability for the eyes. Excess Zn in the hair of children and adolescents with myopia may be associated with the development and progression of this disease [69].

### 4.6. Other Chronic Conditions

Asthma is a common chronic respiratory disorder with inflammation of the airways. The increase in its prevalence in recent years is related to environmental factors, changes in lifestyle, and genetic tendencies [70]. In 17 patients with well-controlled asthma (6–12 years without coexistent malnutrition, any symptoms and treatment the previous year) significantly lower blood Zn levels and lower blood Zn/Cu ratios were found. No variations between the asthma group and the healthy controls were detected in Cu levels [24]. Zn deficiency may be related to IgE production, and this relationship may increase the risk of asthma [71]. A meta-analysis showed that asthma cases had markedly higher levels of Cu among overall Asians inhabitants and may induce OS and chronic inflammation in asthma cases [72]. Cu levels were increased in moderate cases of asthmatic groups and these levels were higher than the mild group [73]. Furthermore, in 17 patients with CF, an autosomal recessive disorder characterized by genetic mutations in the CF transmembrane conductance regulator (CFTR), (14.8 ± 8 years), although serum Cu and Zn were normal, 94% of CF patients had a high Cu/Zn ratio. Serum copper had a positive association with both the Z-score BMI and bone conduction speed (BCS). Even if the inflammation biomarkers evaluated (ESR and CRP) were normal, the high Cu/Zn ratio should alert us to a condition with a high inflammatory response and could reflect the severity of the Zn deficiency [3].

Series of 42 children and 36 adolescents (9.6 ± 4.8 years) with chronic diseases (malnutrition of unknown origin, syndromic diseases, encephalopathies, hyperlipidemia, insulin-dependent DM, and eating disorders), classified by BMI, proved that the mean serum Cu and Zn and the Zn/Cu ratio were normal. Nevertheless, the Cu/Zn ratio was high and 87% of patients with a Cu/Zn ratio > 1.0 and 5% with a Cu/Zn ratio > 2.0 [2] related to a high inflammatory response and severe bacterial infection [74]. Significantly, serum Cu decreased with age and children under 10 years of age, especially those under 5 years of age, developed higher serum Cu levels. Even though all children with hypocupremia had a normal Cu/Zn ratio, all patients with hypozincemia and hypercupremia had a higher Cu/Zn ratio > 1.00. Liver and malnutrition markers were associated with Cu status markers. A strong and significant association was observed between ESR first with the Cu/Zn ratio and second with serum Cu, which may indicate that the Cu/Zn ratio may be used as an inflammatory marker. Moreover, there were meaningful associations between Cu levels with anthropometric, biochemical, dietary, body densitometry, and body composition indicators [2], as presented in a study carried out in CF patients [3]. As mentioned by Van et al. (2020), considering the antagonistic effect of Zn and Cu, when the ratio of Zn to Cu in serum is close to 1:1, the immune response to infectious agents is more effective [75].

### 4.7. Bone Growth and Development

This essential micronutrient plays a role in regulating bone growth and development [76]. It has a positive effect on the proliferation and function of osteoblasts [14,77]. Significantly low serum Cu and Zn levels were found in a sample of 2412 children under 3 years of age. Age and serum Cu and Zn levels were positively associated with bone mineral density (BMD). Children with normal BMD have higher serum Cu and Zn levels than those with low BMD and the incidence of normal BMD increases as the serum Cu and Zn increases [78]. Studies have suggested that the activation of lysyl oxidase, a Cu-containing oxygenase, induces the formation of collagen and elastin crosslinking [76]. What is more, decreased BMD is associated with many primary and secondary pediatric diseases [79]. Among 78 children with chronic diseases, malnutrition was a risk factor for Cu deficiency, and this could be reflected in their BMD [2]. Associations between BMD and trace elements in children with obesity, childhood Wilson’s disease, childhood burns, and thalassemia have been reported and these studies also proved that the intake of trace element deficiencies can correct BMD [80]. Low BMD during adolescence can result in permanent low BMD in adulthood [81]. A NHANES 2007–2018 study, in a sample of 8224 US adults, indicated that total Cu intake was positively associated with increased BMD and negatively associated with osteoporosis risk [82].

### 4.8. Beta-Thalassemia and Sickle Cell Disease

Interestingly, Cu is indispensable for the synthesis of hemoglobin (Hb) and oxidation of iron (Fe) [14]. The most common clinical symptoms associated with Cu deficiency include anemia, leukopenia, and bone lesions (scorbutic-like bone changes and occipital horn) [83]. Beta-thalassemia major (β-TM) patients require frequent blood transfusions to maintain normal Hb levels and suppress ineffective erythropoiesis. β-TM is a genetic disorder caused by mutations in the β-globin gene that reduce its synthesis. Excess free α-globin in maturing red blood cells leads to hemolysis with subsequent anemia [84]. Sixty children (3–12 years) with transfusion-dependent thalassemia (TDT) showed significantly higher Zn and lower Cu levels and a lower Cu/Zn ratio z-score. Reduced Cu was significantly associated with reduced Hb, mean corpuscular volume, and mean corpuscular hemoglobin. Low serum Cu aggravated anemia in TDT patients by reducing erythropoietin [21]. In TDT patients, the release of large amounts of intracellular Zn into plasma due to hemolysis may be a compensatory mechanism that protects against infection and produces anti-inflammatory and antioxidant effects [66]. The Cu/Zn ratio correlated positively with CRP in these patients with normal CRP, suggesting that this ratio in TDT is independent of inflammatory processes [21].

Additionally, Cu is required for hephaestin, a transmembrane Cu-dependent ferroxidase that transports Fe from intestinal enterocytes into the blood, ensuring Fe absorption from dietary intake. The Cu/Zn SOD reduces OS caused by Fe metabolism [83]. Sickle cell anemia (SCA) is the most common form of sickle cell disease (SCD), a chronic condition characterized by hemolytic anemia (frequent blood transfusions), pain crises, and organ damage [85]. Decreased blood levels of antioxidant trace elements may contribute to the pathophysiology in SCA by promoting OS. In 33 children with SCA (8.5 ± 4.1 years), the serum Cu levels and Cu/Zn ratio were significantly higher and the Zn levels were lower. These findings indicate that the assessment of the Cu/Zn ratio may serve as a useful tool for gauging OS [29]. Therefore, 87 patients with SCA (3–15 years old) had significantly lower serum Zn and higher serum Cu levels [86]. In 100 patients with SCA (4–20 years), serum Zn levels were lower and the serum Cu and Cu/Zn ratio were higher than the controls. Significantly higher Cu and lower Zn levels were observed in patients in vaso-occlusive crisis than in those in steady clinical state. The Cu/Zn ratio may be an indicator of disease severity in SCA patients. This finding indicates that demand for Zn utilization in SCD patients increases with disease severity [31].

### 4.9. Neurodevelopment

Copper is critical to neurodevelopment, and its dysregulation in early life has been documented in autism spectrum disorder (ASD) [87]. ASD is a neurodevelopmental condition of heterogeneous etiology [88], and characterized by significant impairments in social interaction, verbal and nonverbal communication, as well as restrictive, repetitive, and stereotypic patterns of behavior [25]. The evidence indicates a greater vulnerability in these children due to a possible alteration of the Zn/Cu cycles [88]. A recent twin study examining deciduous teeth suggested that disrupted dynamics of prenatal Zn–Cu cycles could differentiate ASD and control cases with a 90% accuracy [87]. Although 38 children with a severe form of infantile autism (3.6–18 y) had a Zn level in a normal range, the Cu and CP levels were significantly higher. The Cu/Zn ratio at 2.34 (normal range: 0.70–1.66) became significantly elevated for this group. Compared to the controls, the autism group showed a significant increase in oxidative DNA damage in lymphocytes, plasma CP and Cu levels with a high Cu/Zn ratio, thiol proteins, and SOD activity [27]. Neurodevelopmental regression (NDR) is a subtype of ASD that manifests as a loss of previously acquired developmental milestones. In 27 ASD cases (13 with NDR), children with ASD demonstrated lower prenatal and postnatal Cu and Cu/Zn ratios (corresponding to a high Zn/Cu ratio). Language and communication scores in children with ASD were positively related to prenatal Cu exposure and Cu/Zn ratio [89].

It is important to point out that a study based on 20 children with ASD (6.6 ± 2.9 years), a decreased plasma Zn, higher serum Cu, and lower Zn/Cu ratio were observed. There was a decrease with an increasing severity of ASD as defined by the Childhood Autism Rating Scales score, which suggests that Zn/Cu could be used as a biomarker for the diagnosis of autism. The optimal cutoff value of serum Zn/Cu ratio as an indicator for an auxiliary diagnosis of autism was projected to be 0.81 (85% sensitivity and specificity), with a cutoff of the lowest 2.5% of healthy children. These children appear to be at risk for Zn deficiency, Cu toxicity, and often have low Zn/Cu ratio [25]. In contrast, 25 children with ASD (age 5.96 ± 1.40 years) presented a significant reduction in Zn and Cu levels and Zn/Cu ratio [23]. In 45 ASD children (4–13 years), the Zn/Cu ratio significantly decreased with increased ASD severity, and a significant decrease in Zn/Cu ratio in ASD cases when compared to both children with learning disabilities and the control group was found [90]. Furthermore, in 113 children divided into preschool age (2–5 years old) and school (6–10 years old) age, children had lower whole blood Zn levels and no significant differences in the whole blood Cu and Zn/Cu ratio. Additionally, a significant 6% fall in whole blood Zn levels was detected in preschool-aged children. The whole blood Zn level and Zn/Cu ratio were significantly increased in school-aged children as compared to preschool-aged children in both ASD and control groups [20].

Robberecht et al., in a critical review of the literature on the relation between Zn and Cu and attention deficit hyperactivity disorder (ADHD), suggested that nearly all studies found lower daily Zn intake, in several geographical areas lower Zn levels in ADHD children, and in several meta-analyses a significant association between each other [91]. Skalny et al. and Tippairote et al. examined hair Zn levels and discovered that they were significantly higher in the ADHD group [92,93]. In 20 children (6–16 years) with ADHD, there were lower levels of Zn and Cu compared to both laboratory reference ranges and to normal controls in both hair and serum. There was a significant positive correlation between Zn levels and the inattention, hyperactivity, and impulsivity components of The Conner Score. Therefore, children with low Zn levels may be at increased risk for ADHD [94]. Moreover, in 68 ADHD children (4–9 years), serum Zn levels were 7% lower, serum Cu concentration increased, and the Cu/Zn ratio significantly exceeded control values by 11% [22]. In an intent-to-treat analyses from a randomized controlled trial carried out in 71 ADHD children (7–12 years), there was a substantial 15% Cu deficiency and 6% Cu excess pre-treatment. After 10 weeks with a micronutrient formula (Daily Essential Nutrients), there were no significant differences in Cu, Zn, and Cu/Zn ratio. The following suggests that if children with ADHD and multiple comorbidities do not respond well to established ADHD medications, micronutrients may have an important therapeutic role [26].

We must remember that 63.2% of the studies reported a significantly higher Cu/Zn ratio and 57.9% of them significantly low Zn levels, which correlates with OS, inflammation, malnutrition, depressed immune function [95], and Zn deficiency [16,74]. Cu, like Zn, exhibits a regulatory role in relation to the antioxidant processes that form Cu/Zn SOD, to protect the cell from oxygen radicals [32], which requires a close balance between these two micronutrients. Reduced serum Cu levels decrease Cu/Zn superoxide dismutase activity [32]. High Zn concentrations can inhibit the mitochondrial citric acid cycle and electron transport chain (ETC) enzymes [96]. Cu is essential for the mitochondrial ETC IV complex, cytochrome oxidase. In fact, cells contain specific machinery to transport Cu to the intermembrane space of the mitochondria [97]. Excess Cu can result in toxicity, especially in the brain [98]. In fact, low Cu concentrations may limit the ability of mitochondria to produce ATP, resulting in a background of mitochondrial dysfunction in children with regressive ASD [89].

The main objective was achieved. In this literature review, 63% of the 19 studies show a significantly higher Cu/Zn and Zn/Cu ratio, and 58% significantly lower Zn levels, which suggests that it is essential to consider the evaluation of this biomarker (Cu/Zn or Zn/Cu ratio) routinely in these age groups. Not only does it reveal the oxidative state, but it can also be used as a predictor of the infectious, inflammatory, and nutritional state (risk of Zn deficiency, Cu toxicity) and the commitment at the cellular level of the different tissues involved in Cu metabolism.

## 5. Conclusions

This narrative review of 19 studies of the last five years on the Cu/Zn or Zn/Cu ratios in children and adolescents with acute infections and chronic diseases reveals that 63% of them had a meaningfully higher Cu/Zn and Zn/Cu ratio, and 58% of them a significantly lower Zn level. These groups of patients had a high risk of Zn deficiency, Cu toxicity, a higher risk of developing altered nutritional status, and an infectious and inflammatory state with a tendency to become chronic.

## Figures and Tables

**Table 1 metabolites-13-00082-t001:** Description of copper and zinc concentrations, and the copper/zinc and zinc/copper ratios in children and adolescents (N ° = 19 studies).

Chronic Conditions	Study Design	Year	N°	Age Years	Serum Cu μg/dL		Serum Zinc μg/dL		Serum Cu/Zinc Ratio		Serum Zinc/Cu Ratio	
					Cases	Control	Cases	Control	Cases	Control	Cases	Control
Children												
Infected term/preterm [17] (plasma)	CCS	2017	21	neonates	52.3	-	164.2 ± 43.8	-	0.48 *	0.28	-	-
Small for gestational age (cord blood) [18]	CCS	2021	41	neonates	32.6 ± 14.0	30.3 ± 10.3	61.8 ± 2.4 *	75.6 ± 1.3	0.59 ± 0.061 *	0.43 ± 0.06	-	-
Malaria-infected children [19]	CSS	2018	200	0.5–11	104.2 ± 2.8 **	95.7 ± 2.8	93.4 ± 2.8 **	105.6 ± 2.7	1.16 ± 0.02 **	0.90 ± 0.02	-	-
Autism spectrum disorder [20] ^‡^	CCS	2019	113	2–5	19.9 ± 5.1	20.9 ± 4.4	80.2 ± 13.4	84.9 ± 12.1	-	-	8.10 ± 2.71	8.68 ± 2.74
Transfusion-dependent thalassemia [21]	CCS	2021	60	3–12	69.3 ± 3.5 *	80.8 ± 4.9	57.3 ± 4.2 **	78.2 ± 3.0	−0.28 ± 0.12 ¥,**	0.55 ± 0.17 ¥	-	-
ADHD [22] ∞	CCS	2020	68	4–9	1.22 ± 0.2	1.18 ± 0.2	0.93 ± 0.10 *	1.007 ± 0.17 *	1.32 ± 0.27 **	1.19 ± 0.22	-	-
Autism spectrum disorder [23] (plasma)	CCS	2021	25	5.9 ± 1.4	55.3 ± 22 *	92.3 ± 44.6	222.3 ± 63.8 **	438.5 ± 185.5	2.34	-	4.32 ± 1.02 *	4.88 ± 0.94
Metabolic syndrome [1] (whole blood) #	CSS	2021	911	6–12	1.01 ± 0.13	-	5.21 ± 1.07	-	0.20 ± 0.05	-	-	-
Asthma [24] (blood)	CCS	2021	17	6–12	113 ± 2	105 ± 2	504 ± 106 *	586 ± 116	-	-	4.54 ± 0.92 *	5.64 ± 1.03
Autism spectrum disorder [25] ‽	CCS	2017	20	6.6 ± 2.9	151.6 ± 54.6 *	105.4 ± 16.1	68.7 ± 26.4 *	94.7 ± 11.9	-	-	0.62 ± 0.2 *	0.9 ± 0.09
ADHD pre/post DEN treatment [26] ^‡^	RTS	2019	71	7–12	16.0 ± 3.4	15.5 ± 3.2	12.5 ± 1.8	12.6 ± 1.6	1.3 ± 0.3	1.2 ± 0.2	-	-
Severe infantile autism [27]	CCS	2020	38	7.25 ± 3.9	113 ± 3 *	99 ± 25	84 ± 16	84 ± 11	1.34 ± 0.35 *	1.17 ± 0.23	-	-
Congenital heart disease [28] (blood) #	CCS	2021	39	8.2 ± 1.8	1.1 ± 0.2	1.1 ± 0.1	5.7 ± 1.2	5.9 ± 1.3	-	-	5.26 ± 0.73	5.28 ± 0.99
Sickle cell anemia [29] °	CSS	2019	33	8.5 ± 4.1	130 **	88	61 **	94	1.92	0.98	-	-
Overweight-obese/healthy children [30]	CSS	2020	69	10.9 ± 1.9	109.9 ± 47.9 **	206.4 ± 100.7	85.2 ± 40.6 **	152.9 ± 79.7	1.32 ± 0.33	1.51 ± 0.85	-	-
Children and adolescents												
A series of chronic diseases [2]	CSS	2021	78	1–19	118 ± 29	-	87 ± 12	-	1.4 ± 0.4	-	0.8 ± 0.6	-
Obese/eutrophic patients					119 ± 23	122 ± 3	87 ± 12	88 ± 13	1.4 ± 0.2	1.4 ± 0.4	0.7 ± 1.4	0.8 ± 0.2
Undernutrition/eutrophic patients					114 ± 35	122 ± 3	85 ± 13	88 ± 13	1.4 ± 0.2	1.4 ± 0.4	0.9 ± 0.9	0.8 ± 0.2
Sickle cell anemia [31]	CCS	2019	100	4–20	112.1 ± 2.4 **	102.6 ± 1.6	40.5 ± 1.8 **	54.6 ± 1.2	3.35 ± 0.16 **	1.93 ± 0.05	-	-
In steady clinical state			74		105.8 ± 2.5	102.6 ± 1.6	46.3 ± 1.9 **	54.6 ± 1.2	2.57 ± 0.107 **	1.94 ± 0.052	-	-
In painful crisis/in steady clinical state			26		131.1 ± 4.3 **	105.8 ± 2.5	24.1 ± 0.9 **	46.3 ± 1.9	5.59 ± 0.249 **	2.57 ± 0.107	-	-
Adolescents												
Myopic patients [32]	CCS	2017	83	14.36 ± 2.49	95.6 ± 28.2	92.6 ± 18.3	86.5 ± 22 **	105.4 ± 17.4	0.99 ± 0.20 **	1.196 ± 0.45	-	-
Cystic fibrosis [3]	CSS	2020	17	14.8 ± 8	113 ± 23.5	-	87.2 ± 16.7	-	1.32 ± 0.28	-	0.79 ± 0.18	-

* Legend—^‡^: µmol/L. ∞: µmol/mL. ‽: plasma Zn serum Cu. #: mgL^−1^. °: do not report deviation standard. ADHD: attention deficit hyperactivity disorder. DEN: daily essential nutrients. CCS: case–control study. CSS: cross-sectional study. RTS: randomized trial study. * *p*-value < 0.05. ** *p*-value < 0.01. ¥ zCu_zZn: z-unit-weighted composite score reflecting Copper/Zinc as z copper (zCu)—z zinc (zCu-zZn) score [21].

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
