# Peer review of "Copper/Zinc Ratio in Childhood and Adolescence: A Review"

_metabolites, 2023, doi:10.3390/metabo13010082_

Round 1

Reviewer 1 Report

In this review article, the authors summarized and synthesized the advances in knowledge of the last five years on the Cu/Zn or Zn/Cu ratio in children and adolescents. The authors concluded that children and adolescents with acute and chronic conditions were at greater risk of developing elevated Copper/Zinc ratios, related to altered nutritional, infectious, and inflammatory status.

Comments

This is an interesting review article. The reviewer has some concerns as follows:

1. There are too many abbreviations in the manuscript and it becomes confusing. It is recommended to list these abbreviations in a table.

2. How are children and adolescents defined or distinguished in this manuscript? What are the age ranges for children and adolescents?

3. In Table 1, the data presentation has some confusing. Can data from children and adolescents be compared separately? How about the data from adults? The adults’ data can be an important reference.

4. In Table 1, for Sickle Cell Anemia [79], 2019, some data only have a single number (no standard errors), how to statistical analysis? Moreover, in the Table, what does it mean for the symbol “¥”?

5. The format of Table 1 can be adjusted better.

6. In line 68, it seems to have a truncation at the end of the sentence.

7. In lines 445-450, the descriptions are some confusing. This paragraph is suggested to be rephrased. Moreover, what does it mean for acute and chronic diseases?

8. In Table 1, there is only chronic conditions. How about acute conditions?

Author Response

Changes introduced in the manuscript metabolites-2100468

We acknowledge the reviewers for their useful comments, which helped to improve the quality of the manuscript.

  • Comment 1. This is an interesting review article. The reviewer has some concerns as follows:

Author’s Response: Thank you for your comment.

  • Comment 2. There are too many abbreviations in the manuscript, and it becomes confusing. It is recommended to list these abbreviations in a table.

Author’s Response: Thank you very much for your suggestion. We have reduced the number of abbreviations and the necessary ones have been listed in a table at the end of the manuscript (Line 477).

  • Comment 3. How are children and adolescents defined or distinguished in this manuscript? What is the age ranges for children and adolescents?

Author’s Response: In the material and method section, we have added: “age range: children (≤ 10 years) and adolescents (10-19 years) according to Tanner’s sexual development” (Lines 86-87).

  • Comment 4a. In Table 1, the data presentation has some confusing. Can data from children and adolescents be compared separately?

Author’s Response: Thank you for your suggestion. In Table 1, we show the studies according to whether they were conducted in children, children and adolescents, and adolescents.

  • Comment 4b. How about the data from adults? The adults’ data can be an important reference.

Author’s Response: Thank you for your comment. In the introduction section, we add that the studies described were conducted in adults as well as the elderly (Lines 59-64).

  • Comment 5. In Table 1, for Sickle Cell Anemia [79], 2019, some data only have a single number (no standard errors), how to statistical analysis? Moreover, in the Table, what does it mean for the symbol “¥”?

Author’s Response: Thank you for your comment. In the legend of Table 1, we have explained that the authors did not report standard deviation, and for “¥", we have improved the meaning and written the reference (Lines 100-103).

  • Comment 6. The format of Table 1 can be adjusted better.

Author’s Response: Thank you for your comment. The adjustment has been made.

  • Comment 7. In line 68, it seems to have a truncation at the end of the sentence.

Author’s Response: Thank you for your comment. The error has been fixed.

  • Comment 8. In lines 445-450, the descriptions are some confusing. This paragraph is suggested to be rephrased. Moreover, what does it mean for acute and chronic diseases?

Author’s Response: Thank you for your comment. The text has been changed as follows: “The main aim was achieved. In this literature review, 63% of the 19 studies show a significantly higher Cu/Zn and Zn/Cu ratio, and 58% significantly lower Zn levels, which suggests that it is essential to consider the evaluation of this biomarker (Cu/Zn or Zn/Cu ratio) routinely in these age groups. Not only does it reveal the oxidative state, but it also can use as a predictor of the infectious, inflammatory, and nutritional state (risk of Zn deficiency, Cu toxicity) and the commitment at the cellular level of the different tissues involved in Cu metabolism.” (Lines 457-462).

  • Comment 9. In Table 1, there is only chronic conditions. How about acute conditions?

Author’s Response: Thank you for your comment. In order to better visualize acute and chronic conditions, the discussion section has been divided into subsections: “4.2. Copper to Zinc ratio in acute infections”.

 Thank you very much for your contribution, we really appreciate the review made that enriches the presentation of this research work.

Reviewer 2 Report

The manuscript “Copper/Zinc Ratio in Childhood and Adolescence: A Review” fits the journal’s scope. The authors present a literature short review of the last five years in order to reveal the impact of Cu/Zn ration in children and adolescents. The manuscript is well written, and well-organized. The methods are presented in sufficient detail, with the inclusion and exclusion criteria well described. The data are presented concise and discussed critically. The majority of the references are up to date. Before publication, minor corrections are needed (please see them below).

Lines 82- please correct the error(s)

The discussion section should be divided several subsections. For example, some paragraphs between lines 133-311 represent an introduction. The authors could re-arrange the information in another sub-section.

The conclusion is very short and general. Please consider to extend it.  

Author Response

Changes introduced in the manuscript metabolites-2100468

We acknowledge the reviewers for their useful comments, which helped to improve the quality of the manuscript.

  • Comment 1. The manuscript “Copper/Zinc Ratio in Childhood and Adolescence: A Review” fits the journal’s scope. The authors present a literature short review of the last five years in order to reveal the impact of Cu/Zn ration in children and adolescents. The manuscript is well written, and well-organized. The methods are presented in sufficient detail, with the inclusion and exclusion criteria well described. The data are presented concise and discussed critically. The majority of the references are up to date. Before publication, minor corrections are needed (please see them below).

Author’s Response: Thank you for your comment.

  • Comment 2. Lines 82- please correct the error(s).

Author’s Response: Thank you for your comment. The errors have been fixed.

  • Comment 3. The discussion section should be divided several subsections. For example, some paragraphs between lines 133-311 represent an introduction. The authors could re-arrange the information in another sub-section.

Author’s Response: Thank you very much for your suggestion. The discussion section has been divided into several subsections according to the relationship of the diseases to better visualize the information.

  • Comment 4. The conclusion is very short and general. Please consider extending it. 

Author’s Response: Thank you for your comment. The text has been changed as follows: “This narrative review of 19 studies of the last five years on the Cu/Zn or Zn/Cu ratios in children and adolescents with acute infections and chronic diseases reveals that 63% had a meaningful higher Cu/Zn and Zn/Cu ratio, and 58% a significantly lower Zn level. These groups of patients had a high risk of Zn deficiency, Cu toxicity, a higher risk of developing altered nutritional status, and an infectious and inflammatory state with a tendency to become chronic.”

 Thank you very much for your contribution, we really appreciate the review made that enriches the presentation of this research work.

Round 2

Reviewer 1 Report

This revised manuscript can be accepted. No further comments.